# Eco-Friendly Synthesis of Quinazoline Derivatives Through Visible Light-Driven Photocatalysis Using Curcumin-Sensitized Titanium Dioxide

**DOI:** 10.3390/ma17246235

**Published:** 2024-12-20

**Authors:** Mshari A. Alotaibi, Abdulrahman I. Alharthi, Talal F. Qahtan, Md. Afroz Bakht

**Affiliations:** 1Department of Chemistry, College of Science and Humanities in Al-Kharj, Prince Sattam Bin Abdulaziz University, P.O. Box 173, Al-Kharj 11942, Saudi Arabia; a.alharthi@psau.edu.sa (A.I.A.); m.bakht@psau.edu.sa (M.A.B.); 2Department of Physics, College of Science and Humanities in Al-Kharj, Prince Sattam Bin Abdulaziz University, P.O. Box 173, Al-Kharj 11942, Saudi Arabia; t.qahtan@psau.edu.sa

**Keywords:** sustainable chemistry, visible light photocatalysis, quinazoline derivatives, TiO_2_ nanoparticles, curcumin dye sensitizer, synthetic organic chemistry

## Abstract

This study explores a sustainable method for synthesizing quinazoline derivatives through visible light-driven photocatalysis using curcumin-sensitized titanium dioxide (TiO_2_) nanoparticles. A one-pot, three-component reaction involving aldehydes, urea/thiourea, and dimedone was utilized to efficiently produce quinazoline compounds. The photocatalytic performance of curcumin-sensitized TiO_2_ (Cur-TiO_2_) was compared to pure TiO_2_ (P-TiO_2_), with Cur-TiO_2_ showing significantly enhanced activity. Under optimized conditions—light intensity of 100 mW/cm^2^, catalyst concentration of 1 mg/mL, and a reaction time of 40 min—a 97% product yield was achieved. The Cur-TiO_2_ catalyst demonstrated excellent reusability, maintaining high efficiency over four consecutive cycles with minimal performance loss. This work underscores the potential of natural dye sensitization to extend light absorption of TiO_2_ into the visible spectrum, providing an eco-friendly and cost-effective approach to sustainable organic synthesis.

## 1. Introduction

Quinazoline derivatives, characterized by their unique structure combining a quinazolinone core and an octahydrocyclopenta pyrrole ring system, are highly regarded for their diverse pharmacological and biological activities [1,2,3]. These compounds play pivotal roles in drug discovery, medicinal chemistry, materials science, and the development of synthetic methodologies [4,5,6,7,8]. Traditionally, their synthesis involves multi-step processes, including condensation reactions between amino acids or their derivatives and aldehydes or ketones, followed by cyclization, oxidation, or reduction to form the quinazoline scaffold [9,10,11]. While effective, these conventional methods often require harsh conditions and generate considerable waste. Recent advancements in sustainable chemistry have introduced greener and more efficient synthetic routes for quinazoline derivatives.

Visible light has emerged as a renewable energy resource for catalyzing organic synthesis, providing a sustainable alternative to traditional methods. Visible-light photocatalysis utilizes light-responsive photocatalysts, such as titanium dioxide (TiO_2_) nanoparticles, to drive chemical transformations under mild conditions, reducing the need for toxic reagents and minimizing environmental impact [12,13]. TiO_2_, known for its cost-effectiveness, non-toxicity, and exceptional photochemical properties, has become a prominent photocatalyst [14,15]. However, its utility is limited by its UV-specific absorption and rapid electron-hole recombination. To overcome these challenges, researchers have explored dye sensitization to enhance the functionality of TiO_2_ for use in the visible light spectrum [16,17,18].

The concept of dye-sensitized semiconductors gained prominence with the development of dye-sensitized solar cells (DSSCs) by Grätzel and O’Regan in 1991, demonstrating how dye molecules can broaden the absorption range of semiconductors like TiO_2_ [16]. This breakthrough has spurred innovations in dye design and semiconductor modifications, enabling applications in photocatalysis, pollutant degradation, and organic synthesis [17,19]. Dye molecules, when anchored onto TiO_2_, absorb visible light, become excited, and inject high-energy electrons into the semiconductor’s conduction band. These electrons facilitate redox reactions, including pollutant degradation, water splitting, and organic transformations, while the dye regenerates via interaction with a hole scavenger or redox mediator [20,21].

Natural dyes, such as curcumin, anthocyanins, and chlorophyll, have emerged as eco-friendly alternatives to synthetic dyes [22]. Curcumin, with its broad absorption spectrum of 420–525 nm, is a particularly promising sensitizer due to its alignment with green chemistry principles [23,24,25]. Studies by Goulart et al. [26] and Chawraba et al. [27] have demonstrated the ability of natural dyes to enhance TiO_2_ photocatalysis, providing sustainable solutions for environmental remediation and synthetic applications. Combining renewable dyes with the photocatalytic properties of TiO_2_ offers a viable pathway for eco-friendly chemical production under visible light [28].

This study focuses on synthesizing quinazoline derivatives using a simple, green, one-pot, three-component reaction involving dimedone, urea/thiourea, and aldehydes under visible light. The reaction employs curcumin dye-sensitized TiO_2_ nanoparticles, chosen for their efficient light absorption and energy transfer capabilities. The curcumin dye is prepared through a straightforward extraction process and used to sensitize TiO_2_ nanoparticles. The resulting Cur dye-TiO_2_ photocatalyst is characterized to establish the relationship between dye loading and photocatalytic performance.

The photocatalytic performance of Cur dye-TiO_2_ was evaluated by comparing it with pristine TiO_2_ nanoparticles under various reaction conditions. The study proposes a credible pathway for the photocatalytic process based on the identification of intermediate and reaction products. Additionally, the reusability of Cur dye-TiO_2_ was assessed over multiple cycles, demonstrating minimal performance loss and confirming its potential as a recyclable and sustainable photocatalyst for quinazoline synthesis. This research advances a greener approach to sustainable chemistry through visible-light photocatalysis, aligning with global efforts to develop environmentally friendly methodologies.

## 2. Method

### 2.1. Materials

The P25 TiO_2_ nanoparticles (composition of 80% anatase and 20% rutile) were provided by Evonik. Dimedone, urea/thiourea, and multiple aldehydes procured from Sigma Aldrich were utilized in the synthesis reactions.

### 2.2. Fabrication of Curcumin Dye (Cur Dye)

Curcumin dye can be prepared through a simple extraction process using turmeric powder, which is readily available in most grocery stores. Dissolve the desired amount of turmeric powder in a glass container using absolute ethanol with continuous stirring by a glass rod. Cover the container and let the mixture sit overnight to allow curcumin to dissolve in the solvent. Filter the mixture using filter paper and a funnel to separate the liquid (containing cur dye) from the solid residue. This step is important to remove any insoluble impurities. Collect the filtered liquid that contains curcumin dye in a separate container. Keep in mind that curcumin dye is sensitive to light and can degrade over time. Therefore, we store the dye in a dark, airtight container to preserve its color and properties.

### 2.3. Fabrication of Curcumin Dye-Sensitized TiO_2_ (Cur Dye-TiO_2_)

The following techniques have been carried out with the aim to sensitize the TiO_2_ nanoparticles with the Cur dye in accordance with Figure 1. An initial step involved combining the filtered curcumin dye solution with the TiO_2_ nanoparticles. The resultant mixture was then stirred until the nanoparticles were uniformly dispersed over the mixture. Next, Treatment with ultrasound waves has been used to further improve nanoparticle dispersion and facilitate the bonding of these particles with the molecules of the Cur dye. The TiO_2_ nanoparticles that had been loaded with absorbed Cur dye were extracted from the solution by centrifugation after the mixture was allowed to stand for a period of 24 h.

Following this, the nanoparticles underwent a series of washes with deionized water to eliminate any residual Cur dye or other impurities. Additionally, rinsing with ethanol was performed to eradicate any remaining color molecules or contaminants. Finally, the Cur dye-TiO_2_ sample was thoroughly washed and subsequently dried in an oven at 80 °C for 24 h. To ensure comparability between samples, P-TiO_2_ underwent the same preparation steps as the Cur dye-TiO_2_, except for curcumin dye addition.

### 2.4. Photocatalytic Synthesis of Quinazoline Derivatives

The photocatalytic synthesis of quinazoline (**4a**–**f**) was initiated by the utilization of visible light following the synthesis of nanoparticles made from Cur dye and TiO_2_. The detailed procedure is illustrated in Figure 1. Dimedone (**1**) of 1 mmol, urea/thiourea (**2**) 1 mmol, aldehydes (**3**) 1 mmol, and Cur dye-TiO_2_ photocatalyst (10 mg dispersed in 10 mL ethanol, 1 mg/mL) were all included in the mixture. They mixed the stuff together. There was an improvement in the interaction between the reactant molecules and the TiO_2_ nanoparticles by the utilization of ultrasonic treatment. After that, the entire mixture was subjected to a photocatalytic reaction for a predetermined amount of time while being exposed to visible light through the utilization of a solar simulator which was equipped with a UV-cut-off filter. After the reaction, the mixture undertook filtration to remove any solid particles or debris that may have been present. The solution that was obtained was then extracted using dichloromethane. After that, the organic layer had been separated out, allowed to dry, and examined to confirm the structure of the quinazoline derivatives that were synthesized.

#### Spectroscopic Data of Synthesized Compounds

4-(4-chlorophenyl)-7,7-dimethyl-1,2,3,4,5,6,7,8-octahydroquinazoline-2,5-dione (**4a**)

White solid; FT-IR (cm^−1^, ATR); 3436 and 3330 (NH), 1578 (C=O, ring), 1466 (C=O, urea), 1369 (C=C); ^1^H NMR (DMSO-d6, 400 MHz): 6.96–7.19 (m, 4H, Ar-H), 5.92 (1H, s, CH), 2.51 (s, 2H, CH_2_), 2.33 (s, 2H, CH_2_), 1.03 (s, 6H, 2 × CH_3_); ^13^C NMR (DMSO-d6, 100 MHz): δ 187.65, 155.75, 135.08, 129.84, 129.71, 128.80, 128.21, 127.98, 126.53, 126.43, 123.07, 119.07, 114.75, 109.14, 47.11, 31.74, 28.25.

4-(4-fluorophenyl)-7,7-dimethyl-1,2,3,4,5,6,7,8-octahydroquinazoline-2,5-dione (**4b**)

White solid; FT-IR (cm^−1^, ATR); 3436 and 2955 (NH), 1580 (C=O, ring), 1504 (C=O, urea), 1367 (C=C); ^1^H NMR (DMSO-d6, 400 MHz): 6.75–7.20 (m, 4H, Ar-H), 5.91 (1H, s, CH), 2.50 (s, 2H, CH_2_), 2.28–2.38 (s, 2H, CH_2_), 1.04 (s, 6H, 2 × CH_3_); ^13^C NMR (DMSO-d6, 100 MHz): δ 187.68, 167.23, 159.41, 140.82, 137.38, 128.61, 128.53, 114.99, 110.17, 101.53, 59.36, 57.91, 47.06, 31.74, 28.24.

4-(4-hydroxy-3-methoxy)-7,7-dimethyl-1,2,3,4,5,6,7,8-octahydroquinazoline-2,5-dione (**4c**)

White solid; FT-IR (cm^−1^, ATR); 3188 and 2947 (NH), 1581 (C=O, ring), 1475 (C=O, urea), 1370 (C=C); ^1^H NMR (DMSO-d6, 400 MHz): 10.32 (s, 1H, OH), 6.88–6.90 (d, 1H, J = 7.88 Hz, Ar-H), 6.78–6.80 (d, 1H, J = 7.92 Hz, Ar-H), 6.53–6.55 (d, 1H, J = 7.56 Hz, Ar-H), 5.04 (s, 1H, CH), 3.30–3.79 (s, 3H, OCH_3_), 2.50 (s, 2H, CH_2_), 2.09–2.21 (s, 2H, CH_2_), 0.89–1.05 (s, 6H, 2 × CH_3_); ^13^C NMR (DMSO-d6, 100 MHz): δ 196.19, 164.97, 147.19, 139.69, 126.73, 124.17, 120.19, 111.08, 110.13, 56.00, 50.96, 32.06, 32.03, 29.58, 26.78.

4-(4-chlorophenyl)-7,7-dimethyl-5-oxo-1,2,3,4,5,6,7,8-octahydroquinazoline-2-thione (**4d**)

White solid; FT-IR (cm^−1^, ATR); 3431, 2949 (NH), 1651 (C=O, ring), 1467 (C=O, urea), 1358 (C=C); ^1^H NMR (DMSO-d6, 400 MHz): 7.17–7.30 (m, 4H, Ar-H), 4.51 (s, 1H, CH), 2.24–2.55 (s, 2H, CH_2_), 2.06–2.12 (s, 2H, CH_2_), 0.90–1.04 (s, 6H, 2×CH_3_); ^13^C NMR (DMSO-d6, 100 MHz): δ 196.51, 163.54, 143.72, 131.19, 130.37, 128.28, 127.97, 114.46, 50.49, 32.30, 31.42, 29.06, 27.00.

4-(4-fluorophenyl)-7,7-dimethyl-5-oxo-1,2,3,4,5,6,7,8-octahydroquinazoline-2-thione (**4e**)

White solid; FT-IR (cm^−1^, ATR); 3381, 2955 (NH), 1582 (C=O, ring), 1504 (C=O, urea), 1367 (C=C); ^1^H NMR (DMSO-d6, 400 MHz): 6.94–7.04 (m, 4H, Ar-H), 5.92 (s, 1H, CH), 2.32–2.50 (s, 2H, CH_2_), 2.06–2.08 (s, 2H, CH_2_), 0.85–1.04 (s, 6H, 2 × CH_3_); ^13^C NMR (DMSO-d6, 100 MHz): δ 187.67, 184.50,167.23, 159.39, 137.46, 128.61, 114.98, 101.53, 57.97, 50.92, 47.09, 31.74, 28.25.

4-(4-hydroxy-3-methoxy)-7,7-dimethyl-5-oxo-1,2,3,4,5,6,7,8-octahydroquinazoline-2-thione (**4f**)

White solid; FT-IR (cm^−1^, ATR); 3374 and 3163 (NH), 1589 (C=O, ring), 1465 (C=O, urea), 1374 (C=C); ^1^H NMR (DMSO-d6, 400 MHz): 10.26 (s, 1H, OH), 7.24–7.26 (d, 1H, J = 7.96 Hz, Ar-H), 6.90–7.01 (d, 1H, J = 44.12 Hz, Ar-H), 6.52–6.54 (d, 1H, J = 7.56 Hz, Ar-H), 5.04 (s, 1H, CH), 3.78–3.85 (s, 3H, OCH_3_), 2.50 (s, 2H, CH_2_), 2.05–2.21 (s, 2H, CH_2_), 0.89–1.05 (s, 6H, 2 × CH_3_); ^13^C NMR (DMSO-d6, 100 MHz): δ 192.61, 184.48, 164.96, 151.22, 147.18, 139.68, 126.75, 124.17, 118.21, 110.13, 56.01, 50.96, 32.03, 29.58, 26.78.

The detailed NMR analysis of all synthesized compounds can be found in attached Appendix A (NMR data **4a**–**f**).

### 2.5. Characterization

A comprehensive set of analytical techniques was employed to examine the characteristics of P-TiO_2_ and Cur dye-TiO_2_. The study used Fourier transform infrared (FT-IR) with the Thermo Science (iD5 ATR diamond) Nicolet device (Waltham, MA, USA). For structural analysis, Raman spectroscopy was performed using the SENTERRA II Compact Raman Microscope by (Bruker, Billerica, MA, USA), where the sample powders on a glass slide were probed with a 532 nm laser at 2 mW power, capturing spectra at a resolution of 1 cm^−1^. A UV-visible spectrophotometry test was conducted with a Jasco V-570 spectrophotometer (Jasco Corporation, Tokyo, Japan). Thermal behavior was assessed through Thermogravimetric Analysis (TGA) and Differential Thermal Analysis (DTG) using the Netzsch Proteus 70 system (Netzsch Gerätebau GmbH, Selb, Germany) with Proteus software version 7.0. The samples were heated from 25 to 1000 °C at a rate of 10 °C/min. The Brunauer-Emmett-Telle (BET) surface area, pore radius, and pore volume of the catalyst were analyzed through N_2_-physisorption at 77 K using Quantachrome ASiQwin software (version 5.2) (Boynton Beach, FL, USA). International X-ray diffractometer with Cu Kα radiation (λ = 1.5406 Å) was utilized for X-ray diffraction (XRD) analysis, covering a scanning range of 2θ = 10–70° (Rigaku International, Tokyo, Japan) to ascertain the crystal structure and phase purity of the samples. The morphology of the samples was visualized using Scanning Electron Microscopy (SEM) with the FEI Quanta 250 (Hillsboro, OR, USA). X-ray Photoelectron Spectroscopy (XPS) measurement was performed by a Thermo K-Alpha spectrometer (Waltham, MA, USA). It utilized 1486.6 eV working energy and a 400 μm spot size. Charge adjustment was done during analysis. Furthermore, all binding energy estimations were calibrated to the C 1s energy (284.5 eV) as an internal standard. The Bruker-Plus (400 MHz) Nuclear Magnetic Resonance (NMR) spectrometer (Bruker Corporation, Billerica, MA, USA) was utilized to conduct an analysis of the chemical structures of the quinazoline derivatives that were synthesized. The spectrometer was used to record both ^1^H and ^13^C NMR spectra, with tetramethylsilane serving as the internal standard. These diverse methodologies provided detailed insights into the materials’ attributes and behavior.

## 3. Results and Discussion

### 3.1. Characterization of Curcumin Dye-Sensitized TiO_2_ Photocatalysts

This research advances our knowledge of the complex processes underlying the interaction between curcumin dye and TiO_2_ nanoparticles, highlighting its effectiveness in photocatalytically generating quinazoline derivatives. Insights obtained from FTIR are pivotal in understanding the dynamics of reactions triggered using visible light on the surface of Cur dye-TiO_2_, emphasizing the crucial role of employing natural dyes to promote environmentally friendly photocatalysis in the creation of organic compounds. The FTIR examination focused on the chemical structure and interactions among P-TiO_2_ nanoparticles, Cur dye, and the dye-enhanced TiO_2_, as shown in Figure 2a. The analysis of P-TiO_2_ spectra identified unique peaks, with a notable broad peak at 3343.56 cm^−1^ bringing attention to the existence of hydroxyl (-OH) groups and a band near 1639.20 cm^−1^ suggesting the bending vibrations of adsorbed water molecules. Another peak, which was located at around 652 cm^−1^, was indicative of the stretching vibrations of the Ti-O-Ti band.

The examination of Cur dye through FTIR showcased peaks reflecting its constituent functional groups. A significant broad peak at 3361.32 cm^−1^ was linked to phenolic hydroxyl group stretching vibrations, while additional peaks at 2970.27 cm^−1^ and 2885.84 cm^−1^ were associated with sp2 C-H bond stretching. The presence of a band at 1378 cm^−1^ signified the C-O bond elongation in phenol groups, and a distinct band at 1045.19 cm^−1^ identified the C-O stretching in phenyl alkyl ethers. Peaks at 648.20 and 879.31 cm^−1^ were linked to with the bending vibrations of aromatic C–H bonds.

The detection of OH, CH, and C-O functional groups in the Cur dye-TiO_2_ spectra underscored the successful bonding of Cur dye molecules with the TiO_2_ nanoparticles, suggesting these groups play a substantial role in enhancing the effectiveness of photocatalytic reactions on the TiO_2_ surface.

Raman spectroscopy contains insightful information regarding the molecular composition and interactions within Cur dye-TiO_2_, contributing to material characterization. Raman spectra, presented in Figure 2b, corroborate findings from FTIR spectra on P-TiO_2_, Cur dye, and Cur dye-TiO_2_. In the Raman spectrum for the P-TiO_2_, normal vibration modes, notably the prominent mode at 144 cm^−1^, indicate a well-resolved anatase phase. Additional peaks at 640, 520, and 397 cm^−1^ correspond to other vibration modes [29,30]. The Raman spectrum of Cur dye, spanning the 50–900 cm^−1^ range, reveals bending vibrations of phenyl rings (500–600 cm^−1^), out-of-plane bending (below 400 cm^−1^), and lattice modes and structural vibrations (600–700 cm^−1^) [31].

The Raman peaks in Cur dye-TiO_2_ exhibit broadness and slight wavenumber shifts after adding Cur dye, indicating effective Cur dye adsorption onto TiO_2_ nanoparticles. These findings confirm successful TiO_2_ nanoparticle sensitization with Cur dye, laying a strong groundwork for effective photocatalysis in xanthene derivative production using visible light.

XPS provides a detailed view of the chemical and electronic configuration of materials. In Figure 3, the XPS analysis for both P-TiO_2_ and Cur dye-TiO_2_ is shown, encompassing survey scans and detailed spectra for Ti 2p, O 1s, and C 1s levels. These scans in Figure 3a,b verify the elemental composition of titanium, oxygen, and carbon within both samples, matching anticipated outcomes for the Cur dye-TiO_2_ blend. Specifically, Figure 3b highlights an increased carbon presence in Cur dye-TiO_2_ compared to P-TiO_2_ in Figure 3a, suggesting successful dye integration on the TiO_2_ surface.

The XPS spectrum for P-TiO_2_ against Ti 2p nanoparticles, displayed in Figure 3c, shows clear peaks at roughly 459.5 eV and 465.2 eV, indicative of the Ti 2p_3/2_ and Ti 2p_1/2_ orbitals, respectively. The 5.7 eV gap between these peaks is typical of Ti^4+^ in the TiO_2_ structure [32,33]. Peaks representing Ti^3+^ states are also observed, confirming the dual oxidation state presence in the TiO_2_ matrix, aligning with established literature. For Cur dye-TiO_2_, depicted in Figure 3d, the Ti 2p spectrum reveals pronounced Ti^4+^ peaks and less intense Ti^3+^ signals, with an overall shift towards lower binding energies when compared to the pure P-TiO_2_ spectrum. This shift underscores the effective dye sensitization of TiO_2_, laying the groundwork for improved photocatalytic performance using visible light for synthesizing quinazoline derivatives.

The O1s XPS spectrum of P-TiO_2_, presented in Figure 3e, exhibits a prominent peak at approximately 530.6 eV, corresponding to Ti-O bond energies, which confirms the oxide composition of the material. Additionally, a secondary peak at 532.5 eV, attributed to Ti-OH bonding, highlights the presence of hydroxyl groups, further supporting the coexistence of titanium and oxygen in distinct chemical environments [32]. Conversely, in the Cur dye-TiO_2_, the deconvoluted O1s spectrum (Figure 3f) reveals additional peaks at 530.2, 531.6, and 533.4 eV, indicating interactions between the TiO_2_ and dye molecules, showcasing Ti-O-Ti bonds and highlighting C-O and C=O bonding from the curcumin dye.

The spectrum related to C 1s of P-TiO_2_ (Figure 3g), suggests the existence of surface carbon, likely from environmental contaminants [33]. However, this does not detract from the overall analysis of the composition of nanoparticles. The Cur dye-TiO_2_ C 1s spectrum, illustrated in Figure 3h, exposes a variety of chemical environments for carbon and oxygen atoms within the dye, and additional peaks corresponding to C=O, C-O, C-C, and O-C=O bonds. The Cur dye-TiO_2_ XPS results show that the electronic structure has changed significantly because of the dye adsorption. This proves that the dye was able to sensitize TiO_2_ and that it has the potential to improve both photocatalytic and photoelectrochemical performance.

Figure 4 showcases the XRD patterns and SEM images of P-TiO_2_ and Cur dye-TiO_2_, providing key insights into their structural and morphological characteristics, further supported by FTIR, Raman, and XPS analyses. The XRD (Figure 4a) patterns reveal strong peaks corresponding to the anatase (A) and rutile (R) phases of TiO_2_, with anatase being the dominant phase, as evidenced by peaks such as A(101), A(200), and A(204). This mixed-phase composition, approximately 80% anatase and 20% rutile, is essential for effective charge separation, enhancing photocatalytic activity [32,34]. Importantly, the XRD pattern for Cur dye-TiO_2_ retains the crystalline structure of P-TiO_2_, with no additional peaks for curcumin, confirming its presence as an amorphous or molecularly dispersed layer on the TiO_2_ surface. This finding aligns with FTIR results, which identified characteristic vibrational shifts indicative of the molecular adsorption of curcumin, and XPS analysis, which revealed chemical interactions and charge transfer between curcumin and TiO_2_.

The SEM images complement the XRD findings, illustrating distinct morphological differences between the samples. P-TiO_2_ (Figure 4b) exhibits uniform, densely packed nanoparticles with smooth surfaces, reflecting its pristine nature. In contrast, Cur dye-TiO_2_ (Figure 4c) displays aggregated particles with irregular and rough surfaces, a result of curcumin adsorption on the TiO_2_ surface, likely filling some pores. These structural and surface modifications, as highlighted by XRD, SEM, and complementary spectroscopic techniques, demonstrate the successful sensitization of TiO_2_ with curcumin, enabling enhanced photocatalytic activity under visible light.

Thermal stability and decomposition patterns of P-TiO_2_ nanoparticles and Cur dye-TiO_2_ were evaluated using TGA and DTG, as shown in Figure 5. The TGA profile for P-TiO_2_, shown in Figure 5a, demonstrates a sequential weight reduction up to about 500 °C, with initial losses between 40–200 °C attributed to the desorption of water and volatile compounds from the nanoparticle surface. A continuous weight decrease between 200–510 °C is likely due to the elimination of organic residues adsorbed during production or processing. The DTG curve highlights distinct peaks at 78 °C, 184 °C, and 350 °C, corresponding to water evaporation, volatile desorption, and organic decomposition, respectively. Overall, P-TiO_2_ shows minimal weight loss and retains a high residual mass of 98.17%, indicating excellent thermal stability.

The TGA curve of Cur dye-TiO_2_, shown in Figure 5b, reveals two distinct weight loss stages. The first stage (40–220 °C) is attributed to the loss of adsorbed water and volatiles, like P-TiO_2_. The second stage (220–500 °C) corresponds to the decomposition of organic components from dye sensitization. The DTG profile supports these findings, showing endothermic peaks aligned with each weight loss phase, highlighting the influence of dye on the thermal properties of the composite. Specifically, Cur dye-TiO_2_ exhibits an initial weight loss (~1.43%) between 112–220 °C, followed by a more significant loss (~3.12%) between 220–500 °C, with a residual mass of 95.45%, indicating high thermal stability. In comparison, Pure TiO_2_ shows lower weight losses and a higher residual mass (98.17%), reflecting the absence of organic dye. While Cur dye-TiO_2_ undergoes greater weight loss due to dye incorporation, it retains excellent stability above 500 °C.

Figure 5c illustrates the UV-visible (UV-Vis) absorption spectra of P-TiO_2_, Cur dye, and Cur dye-TiO_2_, highlighting their light-absorption properties and bandgap calculations. P-TiO_2_ exhibits strong absorption in the UV region, consistent with its wide intrinsic bandgap of 3.04 eV and nanoscale particle effects, which indicate an enlarged surface area [20,34]. In contrast, the Cur dye spectrum shows broad absorption from 300 to 600 nm, with a distinct peak at 430 nm corresponding to a π–π* transition, confirming the presence of dye and its ability to absorb visible light effectively. For Cur dye-TiO_2_, the absorption edge shifts to 461 nm, significantly extending into the visible range compared to the 408 nm edge of pristine P-TiO_2_. This red shift indicates the successful sensitization of TiO_2_ by Cur dye, enhancing its capacity to harness visible light [23].

The bandgap energies (Eg) for Cur dye, P-TiO_2_, and Cur dye-TiO_2_ were calculated from their respective absorption edge wavelengths (λ _edge_) using the equation Eg (eV) = 1240/λ_edge_ (nm). The calculated bandgap energies are 2.19 eV, 3.04 eV, and 2.69 eV, respectively, consistent with the findings of Buddee et al. [23].These results underscore the role of Cur dye in narrowing the bandgap of TiO_2_, modifying its electronic structure, and extending its light absorption into the visible spectrum. Such enhancements significantly improve the photocatalytic potential of the composite, making it highly effective for visible-light-driven applications, such as organic synthesis.

The BET results, presented in Table 1, show comparable surface areas for P-TiO_2_ (50.41 m^2^/g) and Cur dye-TiO_2_ (48.23 m^2^/g), indicating that curcumin loading did not significantly affect the overall surface area. However, the reduction in pore volume for Cur dye-TiO_2_ suggests that curcumin molecules occupy and partially block the pores, a finding consistent with the morphological changes observed in the SEM images (Figure 4). The SEM analysis revealed aggregated and irregular particles in Cur dye-TiO_2_ compared to the smooth, uniform nanoparticles in P-TiO_2_, which aligns with the pore filling and structural modification inferred from the BET data. Furthermore, the N_2_ adsorption-desorption isotherms and pore size distribution curves (Figure 5d) confirm the mesoporous nature of both materials, with type IV isotherms and H3 hysteresis loops (p/p° > 0.9), supporting previous findings [35]. These results, together with SEM and BET analyses, highlight the structural and surface alterations induced by curcumin loading, which enhance the photocatalytic capabilities of the composite material.

### 3.2. Photocatalytic Performance

The photocatalytic performance of Cur dye-TiO_2_ in synthesizing quinazoline derivatives is strongly correlated with its structural, optical, and surface property modifications as revealed through detailed characterizations. UV-Vis spectroscopy highlighted the extended absorption range of Cur dye-TiO_2_ into the visible spectrum, enhancing light harvesting under visible light illumination. XRD analysis confirmed the retention of the anatase phase, critical for efficient charge separation, while SEM images demonstrated morphological changes due to curcumin adsorption, which likely enhanced interactions between the catalyst and reactants. BET analysis further emphasized the impact of curcumin in altering the pore structure, optimizing the material’s surface properties for better catalytic activity. These enhancements collectively enabled the superior photocatalytic efficiency of Cur dye-TiO_2_, as demonstrated by its ability to achieve high yields under optimized conditions.

In this study, Cur dye-TiO_2_ was employed as a photocatalyst in a multicomponent reaction involving dimedone, urea/thiourea, and various substituted aldehydes to synthesize quinazoline derivatives (**4a**–**f**). The catalyst provided several advantages, including cost-effectiveness, environmental sustainability, and ease of recovery through simple filtration, thereby minimizing acidic waste and environmental impact. To test the adaptability of this approach, various aryl and heteroaryl aldehydes were successfully converted under optimized conditions, using 10 mg of Cur dye-TiO_2_ in 10 mL of ethanol (1 mg/mL) under visible light illumination. This method yielded high returns of quinazoline derivatives via a one-pot synthesis with straightforward product isolation.

To optimize the reaction conditions, model reactions were performed to evaluate the effects of catalyst concentration, reaction duration, and light intensity. These experiments confirmed that Cur dye-TiO_2_ is a highly effective photocatalyst under visible light, achieving excellent yields. The study demonstrates the potential of Cur dye-TiO_2_ as a sustainable and efficient catalyst for visible-light-driven organic synthesis, combining structural and optical innovations with practical advantages for green chemistry applications.

The photocatalytic performance of Cur dye-TiO_2_ in the synthesis of quinazoline derivatives was systematically evaluated by varying the photocatalyst concentration while maintaining constant light intensity. The results, shown in Figure 6a, indicate a clear relationship between photocatalyst concentration and reaction yield. In the absence of Cur dye-TiO_2_ (0 mg/mL), only 20% yield was achieved after 20 min, highlighting the limited photocatalytic activity of visible light alone. When the photocatalyst concentration increased to 1 mg/mL, the yield significantly rose to 97% after 40 min, demonstrating the critical role of Cur dye-TiO_2_ in enhancing the reaction efficiency. However, increasing the concentration beyond 1 mg/mL resulted in reduced yields (e.g., 95% at 2 mg/mL after 60 min, 93% at 3 mg/mL after 90 min, and 88% at 4 mg/mL after 120 min), indicating an optimal concentration of 1 mg/mL for maximum efficiency.

Figure 6b compares the photocatalytic performance of Cur dye-TiO_2_ and unmodified P-TiO_2_ in synthesizing quinazoline derivatives. While both catalysts facilitated the reactions, Cur dye-TiO_2_ consistently outperformed P-TiO_2_ in terms of reaction rate and yield. After 40 min of visible light exposure, Cur dye-TiO_2_ achieved a 97% yield compared to 82% for P-TiO_2_, underscoring the significant enhancement in catalytic efficiency attributed to curcumin sensitization. This improvement is linked to the increased visible light absorption and surface property modifications induced by the dye, as established in earlier characterizations.

Figure 6c highlights the influence of light intensity on photocatalytic efficiency. A clear correlation was observed, with yields increasing from 55% at 20 mW/cm^2^ to 97% at 100 mW/cm^2^, reflecting the dependence of the reaction rate on photon absorption. However, at intensities beyond 100 mW/cm^2^, a slight decrease in efficiency was noted, possibly due to competing processes such as heat generation. These results demonstrate the importance of optimizing light intensity to maximize photon interactions and photocatalytic activity.

Figure 6d compares the photocatalytic and thermocatalytic approaches for quinazoline synthesis under identical temperature and light conditions. Photocatalysis consistently yielded higher yield (%) than thermocatalysis, confirming the superior efficiency of Cur dye-TiO_2_ under visible light. This advantage is attributed to the enhanced light absorption and charge transfer properties of Cur dye-TiO_2_, as evidenced in UV-Vis and XPS analyses. Thermocatalysis, reliant solely on thermal energy, demonstrated lower efficiency, emphasizing the unique benefits of photocatalytic processes.

The effect of aldehyde substituents on the photocatalytic synthesis of quinazoline derivatives was also investigated (Table 2). Substituents with electron-withdrawing or electron-donating groups influenced reaction times and yields. Aldehydes with electron-withdrawing groups generally exhibited faster reaction rates and higher yields (e.g., entries 1, 2, 4, 5), whereas those with electron-donating groups (entries 3, 6) required longer reaction times. These results underscore the adaptability of the method for structurally diverse substrates and its potential for efficient synthesis of quinazoline derivatives under visible-light-driven photocatalysis.

Additionally, the table highlights a notable trend where compounds incorporating urea (entries 1–3 in Table 2) achieved higher yields and faster reaction completions compared to those involving thiourea counterparts (entries 4–6 in Table 2). This observation suggests that the photocatalytic efficiency and the speed of quinazoline derivative formation are markedly influenced by the specific functional groups attached to the aldehydes, likely due to a combination of steric and electronic effects on the reaction dynamics.

These insights, consistent with findings from prior research involving various metal catalysts, underline the adaptability and efficacy of Cur dye-TiO_2_ in facilitating the synthesis of quinazoline derivatives. The study not only showcases the influence of aldehyde substituents on the photocatalytic activity but also points towards optimizing catalyst design for enhanced selectivity and efficiency in similar synthetic pathways. The results from this comprehensive evaluation provide valuable guidance for the improvement of catalyst efficacy and selectivity in the production of quinazoline derivatives and potentially other organic compounds.

#### 3.2.1. Mechanistic Approach

We present a mechanism demonstrating how curcumin dye sensitization of TiO_2_ nanoparticles under visible light catalyzes the efficient synthesis of quinazoline derivatives through a one-pot, three-component condensation reaction, as depicted in Figure 2. When curcumin dye adheres to the surface of TiO_2_ nanoparticles, it extends the absorption spectrum into the visible light region. Acting as a photosensitizer, curcumin absorbs visible light and transfers the energy to TiO_2_, exciting electrons from the valence band to the conduction band. Consequently, electron-hole pairs are generated within the TiO_2_ nanoparticles. The excited electrons in the conduction band exhibit high reducing potential, while the holes in the valence band possess significant oxidizing capabilities. The holes (h^+^) in the valence band can engage with ethanol, serving as a hole scavenger, thus participating in oxidation reactions to minimize charge carrier recombination [20,23]. Simultaneously, photoexcited electrons interact with molecular oxygen (O_2_) adsorbed on the TiO_2_ nanoparticle surface, generating superoxide radical anions (O_2_•−). These reactive species play a crucial role in activating the aldehyde (RCHO), facilitating its subsequent condensation reaction with urea. In this process, the carbonyl group of the aldehyde reacts with the nucleophilic amine group of urea, leading to the formation of an intermediate imine. Dimedone (a cyclic diketone, shown in red) reacts with the intermediate formed from aldehyde and urea. This step likely proceeds via nucleophilic attack, leading to a more complex intermediate structure. Dimedone is commonly involved in multi-component reactions to form fused rings. The condensation product undergoes cyclization, leading to the formation of the quinazoline core structure. This step involves the closure of a ring system, likely facilitated by dehydration (loss of water), forming the fused heterocycle characteristic of quinazoline derivatives. After cyclization, the final product is a quinazoline derivative (green nitrogenous structure). The “X” in the structure indicates variability in substituents (either oxygen or sulfur, based on the image), which can alter the electronic properties and biological activity of the final compound. This proposed mechanism illustrates a sustainable chemistry approach by harnessing visible light, a renewable energy source, to activate TiO_2_ nanoparticles sensitized with natural dye, curcumin. Furthermore, the one-pot synthesis reduces the number of synthetic steps and the requirement for additional reagents, thereby minimizing waste generation and environmental impact.

#### 3.2.2. The Recyclability Studies of Cur Dye-TiO_2_ as Photocatalyst

To investigate the practicality and sustainability of Cur dye-TiO_2_ photocatalyst, we examined its reusability, a criterion vital for decreasing costs and environmental effect. Multiple rounds of photocatalytic synthesis of quinazolines under visible light were performed to explore Cur dye-TiO_2_ reusability. Four cycles of reuse were conducted while maintaining the time of irradiation consistent at 40 min and utilizing an optimized concentration of Cur dye-TiO_2_ (1 mg/mL).

Figure 7 presents the results for the recycling experiments involving the Cur dye-TiO_2_ as a photocatalyst. After each cycle, the used Cur dye-TiO_2_ photocatalyst was retrieved with a remarkable yield (97%) and underwent further drying before reuse. Across three successive recycling stages, the Cur dye-TiO_2_ photocatalyst consistently sustained its performance and yield.

In the initial recycling round, the yield percentage peaked at 96%, remaining robust at 95% and 92% in the subsequent two cycles. These findings underscore the exceptional recycling of Cur dye-TiO_2_, highlighting its capacity to sustain photocatalytic performance over numerous iterations. This research underscores the promise of Cur dye-TiO_2_ as a dependable and effective environmentally friendly photocatalyst production using visible light, further enhanced by its reusability for multiple cycles.

## 4. Conclusions

This work achieves a notable advancement in sustainable chemistry by employing visible light with the photocatalyst dye-TiO_2_ for synthesizing quinazoline derivatives under mild and eco-friendly conditions. After meticulous optimization involving a time of reaction (40-m), an intensity of visible light (100 mW/cm^−2^), and a concentration of photocatalyst (1 mg/mL), we determined the ideal experimental parameters for the photocatalytic production process. Our comparative examination underscored the efficient photocatalysis performance of Cur dye-TiO_2_ over P-TiO_2_ nanoparticles, confirming the significance of sensitization for successful organic reactions. Additionally, our examination of Cur dye-TiO_2_ reusability showcased impressive stability and endurance. The photocatalyst retained consistent activity over four consecutive uses, with minimal decline noted. This capacity for reuse presents notable cost-effectiveness and environmental benefits, rendering it an appealing choice for real-world applications. Overall, this work showcases the effective fusion of a combination of natural dye sensitization and photocatalysis using visible light, connecting green chemistry with synthetic organic chemistry. It advances sustainable chemical synthesis and lays the framework for future developments.

## Data Availability

Data will be provided upon request.

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
