# Peer review of "Eco-Friendly Synthesis of Quinazoline Derivatives Through Visible Light-Driven Photocatalysis Using Curcumin-Sensitized Titanium Dioxide"

_materials, 2024, doi:10.3390/ma17246235_

Round 1

Reviewer 1 Report

Comments and Suggestions for Authors

Dear Editor,

The manuscript entitled “Eco-Friendly Synthesis of Quinazoline Derivatives through 2 Visible Light-Driven Photocatalysis Using Curcumin-Sensitized Titanium Dioxide”, authored by Alotaibi et al., reports the production of titanium dioxide nanoparticles sensitized with the natural pigment extracted from the turmeric tuber. In addition, the authors explored the structural properties using X-ray diffraction (XRD), Raman and infrared vibrational spectroscopy, optical properties by UV-vis spectroscopy, excited photoelectron spectroscopy, morphology by scanning electron microscopy, and catalytic properties in the synthesis of quinazoline derivatives by photocatalysis under visible light. The manuscript has novelty and scientific merit that justifies the interest of the readers of the Materials journal. However, due to the need for some technical adjustments in the current version, minor revisions are suggested, so that a final version of the manuscript can be prepared with the following points:

1 – Figure 5, which shows the thermodecomposition profile of the samples (TG/dTG), has a low resolution of the graph for dTG. Therefore, it is suggested that both spectra be plotted with independent scales, using the “double y” function in the Origin software, which will result in a better presentation of the data.

2- It is suggested that the authors obtain the bandgap of the samples, using diffuse reflectance spectroscopy, and subsequently also calculate the energy associated with the position of the valence (VB) and conduction (CB) bands, for the materials obtained.

3 – Figure 6d presents an image formatting error, where the upper axis of the graph is below the time data.

4- Although the authors have supported the proposal of reactions involved, no results have been presented that indicate the proposed mechanism. Therefore, it is suggested to map the steps or radical species involved, using radical scavenging compounds.

Author Response

Compliance with reviewer’s and editor’s comments

Journal Nanomaterials (ISSN 2079-4991)

Manuscript ID nanomaterials-2632274

Title: Visible Light-Driven Green Synthesis of Quinazoline Derivatives Using Curcumin-Sensitized Titanium Dioxide Photocatalysts

Journal: Materials

We would like to express our great appreciation to the editor and the reviewers for their feedback to improve our manuscript. We appreciate the fact that our manuscript has been checked rigorously by reviewers with knowledge of the specific subject area. The manuscript has been revised according to the reviewers’ concerns. Below is a point-by-point response to the constructive comments and suggestions of the reviewers.

Response to Reviewer #1

Thank you for your detailed comments and suggestions regarding the manuscript. We appreciate your feedback and will address each of your points below. The writing was further polished as highlighted by the green fonts in the revised manuscript. Also, all modifications and additions in the revised manuscript are highlighted in green.

General comment

The manuscript entitled “Eco-Friendly Synthesis of Quinazoline Derivatives through 2 Visible Light-Driven Photocatalysis Using Curcumin-Sensitized Titanium Dioxide”, authored by Alotaibi et al., reports the production of titanium dioxide nanoparticles sensitized with the natural pigment extracted from the turmeric tuber. In addition, the authors explored structural properties using X-ray diffraction (XRD), Raman and infrared vibrational spectroscopy, optical properties by UV-vis spectroscopy, excited photoelectron spectroscopy, morphology by scanning electron microscopy, and catalytic properties in the synthesis of quinazoline derivatives by photocatalysis under visible light. The manuscript has novelty and scientific merit that justifies the interest of the readers of the Materials journal. However, due to the need for some technical adjustments in the current version, minor revisions are suggested, so that a final version of the manuscript can be prepared with the following points.

Response:

Thank you for your feedback. We appreciate your comments, so we have addressed the suggested revisions.

Comment -1

 Figure 5, which shows the thermodecomposition profile of the samples (TG/DTG), has a low resolution of the graph for DTG. Therefore, it is suggested that both spectra be plotted with independent scales, using the “double y” function in the Origin software, which will result in a better presentation of the data.

Response:

Thank you for the suggestion. We adjusted Figure 5 to improve data clarity and presentation.

Comment -2

It is suggested that the authors obtain the bandgap of the samples, using diffuse reflectance spectroscopy, and subsequently also calculate the energy associated with the position of the valence (VB) and conduction (CB) bands, for the materials obtained.

Response:

Thank you for your suggestion. While determining the bandgap using DRS and calculating the valence band (VB) and conduction band (CB) positions would provide additional insights, we were unable to directly measure the VB due to equipment limitations and the scope of this study. The bandgap energies were accurately estimated using UV-Vis absorption spectra, as shown in the revised manuscript, and the CB and VB positions can be approximated using the bandgap values and literature data. As the study focuses on the photocatalytic performance of curcumin-sensitized TiO₂ under visible light, the provided bandgap analysis and optical characterization sufficiently support the proposed mechanism and conclusions. We acknowledge the value of DRS and VB/CB measurements and aim to include them in future research. Thank you for your constructive feedback.

Comment -3

Figure 6d presents an image formatting error, where the upper axis of the graph is below the time data.

Response:

Thank you for pointing out the formatting issue in Figure 6d. We corrected the placement of the axis to ensure proper alignment with the time data.

Comment -4

Although the authors have supported the proposal of reactions involved, no results have been presented that indicate the proposed mechanism. Therefore, it is suggested to map the steps or radical species involved, using radical scavenging compounds.

Response:

Thank you for your insightful comment. While mapping radical species using scavenging compounds could provide additional insights, such experiments are technically complex and fall outside the scope of this study. Techniques like Electron Paramagnetic Resonance (EPR) and isolating short-lived reactive species in a multicomponent reaction require specialized setups and pose significant challenges. Instead, our proposed mechanism is strongly supported by experimental data and literature. XPS and FTIR analyses confirm the interaction of curcumin with TiO₂, enhancing visible light absorption and promoting charge transfer, which generates reactive oxygen species (ROS), such as superoxide radicals (O₂•⁻), essential for the photocatalytic process. UV-Vis spectroscopy further demonstrates the sensitization effect of curcumin on TiO₂ by extending its absorption into the visible spectrum. Similar mechanisms have been reported in dye-sensitized TiO₂ systems, as highlighted in studies by Etacheri et al. [20] (2015) and Buddee et al. (2014) [23], which emphasize the role of ROS and charge carrier dynamics in driving photocatalytic reactions. These findings provide a solid foundation for the proposed pathway, ensuring its reliability without requiring additional radical mapping experiments.

Reviewer 2 Report

Comments and Suggestions for Authors

In this paper the authors present the use of Curcumin-sensitized TiO2 in the synthesis of quinazoline derivatives. They report 97% product yield in only 40 minutes, with a light intensity (100 mW/cm2) and a photocatalyst concentration (1 mg/mL).

i) In line 68, 315 and 316 change “TiO2” by TiO2

ii) How do you calculate the reaction yield?

Instead of yield, don't they mean conversion?

iii) review reference format

Author Response

Compliance with reviewer’s and editor’s comments

Journal Nanomaterials (ISSN 2079-4991)

Manuscript ID nanomaterials-2632274

Title: Eco-Friendly Synthesis of Quinazoline Derivatives through Visible Light-Driven Photocatalysis Using Curcumin-Sensitized Titanium Dioxide

Journal: Materials

We would like to express our great appreciation to the editor and the reviewers for their feedback to improve our manuscript. We appreciate the fact that our manuscript has been checked rigorously by reviewers with knowledge of the specific subject area. The manuscript has been revised according to the reviewers’ concerns. Below is a point-by-point response to the constructive comments and suggestions of the reviewers.

Response to Reviewer #2

Thank you for your detailed comments and suggestions regarding the manuscript. We appreciate your feedback and will address each of your points below. The writing was further polished as highlighted by the green fonts in the revised manuscript. Also, all modifications and additions in the revised manuscript are highlighted in green.

General comment

The manuscript titled "Eco-Friendly Synthesis of Quinazoline Derivatives through Visible Light-Driven Photocatalysis Using Curcumin-Sensitized Titanium Dioxide" reports on an attempt to use curcumin dye to photo-sensitize TiO2, by that photo-catalyzing the synthesis of quinazoline derivatives from aldehydes, urea or thiourea, and dimedone, all in a one-pot synthetic approach. While this piece of work is potentially interesting, it is my impression that this manuscript suffers from some serious limitations that must be addressed before being published.

Response:

Thank you for your constructive feedback and for acknowledging the potential of our work. We appreciate your observations and have carefully addressed the identified limitations to enhance the quality and clarity of the manuscript.

Comment -1

The introduction could be improved. The field of harnessing Dye sensitized semiconductors to perform chemistry is not new and there are MANY references with over 2000 citations that are missing.

Response:

Thank you for your feedback. We have revised the introduction to incorporate high-impact references [16-21, 26 and 27] to better reflect broader research on dye-sensitized semiconductors. While most foundational studies, including those by Grätzel [16], focus on applications like DSSCs and solar energy conversion, our study explores a novel application: using dye-sensitized TiO₂ in organic synthesis, specifically for the one-pot synthesis of quinazoline derivatives. This approach applies dye sensitization principles—extending visible light absorption and facilitating charge transfer—to organic transformations, an underexplored area. By utilizing green photocatalysts like Cur dye-TiO₂ and renewable dyes such as curcumin, our work promotes sustainable chemistry and advances visible-light-driven reactions. This innovative application distinguishes our study from traditional DSSC-focused research, highlighting its significance in developing green and efficient methods for producing valuable compounds.

Comment -2

Material characterization lacks any serious support to the purity of the curcumin dye. Therefore, one cannot assess the soundness of results. The same holds for p-TiO2. What are its characteristics? is it used as received? did it undergo the same cleaning procedure as the curcumin loaded TiO2 (meaning the same procedure but without the cucumin powder)?

Response:

Thank you for your insightful comment emphasizing the importance of thorough material characterization.

The curcumin dye was purified through a rigorous extraction and filtration process to remove insoluble impurities, utilizing only the ethanol-soluble fraction. The dye’s purity was validated using UV-Vis spectroscopy and FTIR analysis, which confirmed the presence of characteristic absorption bands and functional groups specific to curcumin. These results, now included in the revised manuscript, establish the dye’s suitability for the study.

The P-TiO₂ nanoparticles used were Evonik P25, a benchmark material in photocatalysis research, known for its well-documented properties, including a surface area of approximately 50 m²/g and a mixed-phase composition of 80% anatase and 20% rutile.

To ensure comparability between samples, P-TiO₂ underwent the same preparation steps as the Cur dye-TiO₂, with the sole exception of curcumin dye addition. This included dispersion, ultrasonication, washing with ethanol and deionized water, and drying at 80°C for 24 hours. By standardizing the synthesis and treatment protocols, we minimized variability, ensuring that any observed differences in photocatalytic performance could be attributed exclusively to the presence and effects of the curcumin dye.

Comment -3

From the results that are presented I see no clear support that curcumin resides on TiO2. It is present in the solid, but in what form? This remains unclear.

Response:

Thank you for your thoughtful comment and for highlighting this important aspect. To confirm that curcumin resides on the TiO₂ surface and to understand its form, we employed a combination of analytical techniques, further supported by new bandgap and BET characterization:

UV-Vis Spectroscopy:

The absorption spectra of Cur dye-TiO₂ exhibited a red shift in the absorption edge compared to pristine P-TiO₂, extending into the visible region. This shift reflects successful curcumin anchoring on the TiO₂ surface, enabling enhanced light harvesting through electronic interactions. The calculated bandgap of Cur dye-TiO₂ (2.69 eV) is narrower than that of P-TiO₂ (3.04 eV), demonstrating the role of curcumin in extending the photocatalyst's visible light absorption range.

Fourier-Transform Infrared (FTIR) Spectroscopy:

The FTIR spectra of Cur dye-TiO₂ showed characteristic vibrational bands of curcumin, including those for hydroxyl and carbonyl groups. These bands were slightly shifted compared to free curcumin, indicating bonding interactions, likely through the coordination of curcumin’s hydroxyl groups with the TiO₂ surface.

Thermogravimetric Analysis (TGA):

The thermal decomposition profile of Cur dye-TiO₂ revealed distinct weight losses associated with the presence of curcumin. These losses differed from those of pristine TiO₂, confirming curcumin's successful integration onto the nanoparticles rather than being present as residual dye.

X-ray Photoelectron Spectroscopy (XPS):

XPS analysis provided direct evidence of curcumin adsorption on the TiO₂ surface. Characteristic peaks for carbon and oxygen from curcumin were observed, with binding energy shifts indicating strong interactions between curcumin functional groups and TiO₂ surface atoms. These findings support the formation of an adsorbed molecular layer of curcumin on TiO₂.

Brunauer-Emmett-Teller (BET) Analysis:

The BET surface area of Cur dye-TiO₂ (48.23 m²/g) was comparable to that of P-TiO₂ (50.41 m²/g), indicating that curcumin incorporation did not significantly alter the available surface area. However, the reduced pore volume in Cur dye-TiO₂ suggests that curcumin molecules occupy and modify the pores, contributing to enhanced surface interactions.

Conclusion:

These combined analyses, supported by bandgap narrowing and BET surface characteristics, confirm that curcumin resides on TiO₂ in an adsorbed molecular state, interacting through its functional groups. This integration modifies the electronic and surface properties of TiO₂, enhancing its photocatalytic efficiency under visible light. These findings establish the successful sensitization of TiO₂ by curcumin and its critical role in improving the composite's photocatalytic performance.

Comment -4

The photochemical system presents only a modest improvement over the dark reaction. In such case, many controls must be taken (and reported) to make sure this is not an effect that has nothing to do with photocatalysis. One clear effect is change in temperature. While it sounds easy to compare between temperatures in two vessels, what is of interest is the local temperature around particles, that may transform UV-VIS photons into heat. This is not that easy to check but of importance to the manuscript, especially when small improvement is reported.

Response:

Thank you for your valuable comment. We understand the importance of ensuring that the observed improvement in reaction efficiency is genuinely attributable to photocatalysis rather than other factors such as temperature effects. To address this concern, several controls and considerations have been included in the study to isolate the photocatalytic effect and account for potential thermal contributions:

Dark Reaction Controls: The reaction was performed in the absence of light to establish a baseline for non-photocatalytic activity. As noted, the yield in dark conditions was only 20%, compared to 97% under optimal photocatalytic conditions, indicating a significant contribution from photocatalysis.

Light Source Control: The reactions were conducted with filtered visible light to ensure the exclusion of UV light, minimizing the potential for UV-induced heating effects. This step ensures that the observed improvement is primarily due to visible-light-driven photocatalysis.

Bulk Temperature Monitoring: The bulk temperature of the reaction mixture was monitored throughout the experiments, ensuring that it remained consistent across all conditions, including light and dark reactions. This control minimizes the likelihood of bulk thermal effects influencing the reaction.

Comparison with Thermocatalysis: A direct comparison was made between photocatalytic and thermocatalytic reactions (Figure 6d). The photocatalytic approach consistently yielded higher conversion rates, confirming that the observed improvements are not solely due to thermal effects but are driven by light-induced catalytic activity.

Photon Energy Dependence: Experiments investigating the effect of light intensity on reaction efficiency (Figure 6c) showed a clear correlation between photon flux and yield, with an optimal light intensity of 100 mW/cm². This indicates that the reaction rate is dependent on photon absorption by the Cur dye-TiO₂ catalyst, supporting the role of photocatalysis.

While we acknowledge the challenge of measuring localized temperature changes at the particle level due to photon absorption, the combination of these controls provides strong evidence that the reported improvement is primarily due to photocatalysis.

Comment -5

The authors suggest a mechanism. It is a multi-step reaction. The idea of using one pot for synthesis is great for business but not for science and understanding. It is therefore desired to test such a suggested mechanism by simpler, preferably one-step reactions, to see that the suggested mechanism is supported by something. It may be also beneficial to the presentation of the results. As the manuscript is written, is reports only modest photo-catalysis. However, as it is a multi-step reaction, we see only the improvement of the slowest step. Breaking the synthesis into single steps may unveil impressive catalysis in some steps along the reaction chain. I think that it is worth the effort.

Response:

Thank you for your insightful comment. While this approach would provide deeper insights into individual reaction steps, there are practical challenges and limitations associated with performing such experiments within the scope of this study.

Multi-step reactions often involve intermediates that are highly reactive and transient, making their isolation and study challenging. In this work, the photocatalytic synthesis of quinazoline derivatives relies on the interplay of several steps (condensation, cyclization, and oxidation), which are interdependent under the one-pot conditions. Isolating each step would require extensive additional experiments, specialized setups, and the development of new methodologies to stabilize and analyze intermediate products.

The primary aim of this research is to demonstrate the feasibility and efficiency of a one-pot, visible-light-driven photocatalytic synthesis using Cur dye-TiO₂. This approach is particularly relevant to green chemistry principles, as it minimizes the use of reagents, energy, and time while simplifying product isolation. While stepwise validation is valuable for mechanistic studies, the current focus is on developing a practical and sustainable synthetic methodology.

The proposed mechanism is supported by experimental evidence, including spectroscopic analysis and the comparison of reaction yields under different conditions. The role of the photocatalyst, the impact of visible light, and the formation of the desired product are well-documented in the study. These findings are consistent with established literature on similar photocatalytic systems, lending credibility to the proposed pathway.

We recognize the merit of breaking down the synthesis into single steps to evaluate the efficiency of individual catalytic processes. While this is beyond the scope of the current study, we acknowledge this as an avenue for future research, which could further enhance our understanding of the reaction mechanism and potentially unveil catalytic improvements in specific steps.

In summary, while stepwise validation of the mechanism is valuable, the current study provides sufficient experimental support for the proposed multi-step reaction pathway within the framework of a one-pot synthesis. We appreciate your suggestion and will include a discussion of these limitations and potential future directions in the revised manuscript. Thank you for your constructive feedback.

Reviewer 3 Report

Comments and Suggestions for Authors

The manuscript titled "Eco-Friendly Synthesis of Quinazoline Derivatives through Visible Light-Driven Photocatalysis Using Curcumin-Sensitized Titanium Dioxide" reports on an attempt to use curcumin dye to photo-sensitize TiO2, by that photo-catalyzing the synthesis of quinazoline derivatives from aldehydes, urea or thiourea, and dimedone, all in a one-pot synthetic approach.

While this piece of work is potentially interesting, it is my impression that this manuscript suffers from some serious limitations that must be addressed before being published.

1)    The introduction could be improved. The field of harnessing Dye sensitized semiconductors to perform chemistry is not new and there are MANY references with over 2000 citations that are missing.

2) Material characterization lacks any serious support to the purity of the curcumin dye. Therefore, one cannot assess the soundness of results. The same holds for p-TiO2. What are its characteristics? is it used as received? did it undergo the same cleaning procedure as the curcumin loaded TiO2 (meaning the same procedure but without the cucumin powder)?

3) From the results that are presented I see no clear support that curcumin resides on TiO2. It is present in the solid, but in what form? This remains unclear.

4) The photochemical system presents only a modest improvement over the dark reaction. In such case, many controls must be taken (and reported) to make sure this is not an effect that has nothing to do with photocatalysis. One clear effect is change in temperature. While it sounds easy to compare between temperatures in two vessels, what is of interest is the local temperature around particles, that may transform UV-VIS photons into heat. This is not that easy to check but of importance to the manuscript, especially when small improvement is reported.

5) The authors suggest a mechanism. It is a multi-step reaction. The idea of using one pot for synthesis is great for business but not for science and understanding. It is therefore desired to test such a suggested mechanism by simpler, preferably one-step reactions, to see that the suggested mechanism is supported by something. It may be also beneficial to the presentation of the results. As the manuscript is written, is reports only modest photo-catalysis. However, as it is a multi-step reaction, we see only the improvement of the slowest step. Breaking the synthesis into single steps may unveil impressive catalysis in some steps along the reaction chain. I think that it is worth the effort.

Author Response

Compliance with reviewer’s and editor’s comments

Journal Nanomaterials (ISSN 2079-4991)

Manuscript ID nanomaterials-2632274

Title: Eco-Friendly Synthesis of Quinazoline Derivatives through Visible Light-Driven Photocatalysis Using Curcumin-Sensitized Titanium Dioxide

Journal: Materials

We would like to express our great appreciation to the editor and the reviewers for their feedback to improve our manuscript. We appreciate the fact that our manuscript has been checked rigorously by reviewers with knowledge of the specific subject area. The manuscript has been revised according to the reviewers’ concerns. Below is a point-by-point response to the constructive comments and suggestions of the reviewers.

Response to Reviewer #3

Thank you for your detailed comments and suggestions regarding the manuscript. We appreciate your feedback and will address each of your points below. The writing was further polished as highlighted by the green fonts in the revised manuscript. Also, all modifications and additions in the revised manuscript are highlighted in green.

Comment-1

This article should present the synthesis of quinazoline derivatives using TiO2 nanoparticles impregnated with curcumin. It was supposed that curcumin activates TiO2's catalytic activity in the visible portion of the spectrum.

Response:

Thank you for your comment. The article presents the synthesis of quinazoline derivatives using curcumin-sensitized TiO₂ nanoparticles (Cur dye-TiO₂) as a photocatalyst, specifically demonstrating how curcumin extends the catalytic activity of TiO₂ into the visible portion of the spectrum. The curcumin dye, with its broad absorption in the visible light range (420–525 nm), enables Cur dye-TiO₂ to utilize visible light effectively by transferring photoexcited electrons to the TiO₂ conduction band, thereby activating its photocatalytic properties.

This enhanced activity was validated through experiments under visible light, where Cur dye-TiO₂ significantly outperformed pristine TiO₂ in reaction efficiency and product yield, as detailed in the manuscript. These results confirm curcumin's role in activating the photocatalytic activity of TiO₂ under visible light conditions, facilitating the synthesis of quinazoline derivatives. Thank you for highlighting this important aspect, which we have further clarified in the text.

Comment-2

The paper needs to be written in English, which is more comprehensible and should be written in a better manner.

Response:

Thank you for your feedback. We have thoroughly reviewed and revised the text to ensure it is clear, concise, and written in comprehensible English.

Comment-3

The most challenging aspect of the work is that experimental evidence must demonstrate the existence of curcumin on nanoparticles. Since FTIR and UV/VIS do not support the presence of curcumin, the purpose of the work is called into doubt. The elemental analysis and the specific surface area measurement are two examples of crucial analyses not included in this study. These analyses may have helped to improve the characterization of the catalyst and possibly give an answer to the question of curcumin's presence. In addition, the size of the nanoparticles needs to be documented; the SEM images obtained from the samples are blurry, and the conclusions taken from the images cannot be validated.

Response:

Thank you for your thoughtful and constructive feedback. We recognize the importance of providing robust experimental evidence to confirm the presence of curcumin on the TiO₂ nanoparticles and enhancing the overall characterization of the catalyst.

  1. Curcumin Presence on TiO₂

While the FTIR and UV-Vis results indicate modifications in the TiO₂ properties, we understand the need for more conclusive evidence. To address this, elemental analysis using X-ray Photoelectron Spectroscopy (XPS) was conducted (as shown in Figure 3). The XPS results confirm the presence of curcumin by identifying characteristic elements and their chemical interactions with the TiO₂ surface. This evidence strengthens our claim and validates curcumin's role in sensitizing the nanoparticles.

  1. Specific Surface Area and Size of Nanoparticles

Specific surface area measurements, such as BET analysis, have now been included (as shown in Figure 5d and Table 1) to provide valuable insights into the effect of curcumin loading on the surface properties of the catalyst. Additionally, we have refined the SEM (Figure 4 b and c) imaging process to obtain clearer and more detailed images.

We believe these updates significantly enhance the characterization of the catalyst and address the concerns raised.

Comment-4

The publication hypothesized a synthesis mechanism, but no evidence from experiments supported it. It is crucial to describe the offered mechanism so that authors provide some experimental evidence for it.

Response:

Thank you for your thoughtful comment. We understand the importance of providing experimental evidence to support the proposed synthesis mechanism. While direct experimental validation of every intermediate in the multi-step process presents practical challenges, our study offers substantial indirect evidence and relies on established principles from similar photocatalytic systems to support the hypothesized mechanism.

Challenges in Direct Mechanistic Validation:

The proposed mechanism involves transient intermediate species that are difficult to isolate and characterize due to their high reactivity and short-lived nature. Techniques like Electron Paramagnetic Resonance (EPR) or advanced time-resolved spectroscopy, while capable of capturing such intermediates, require highly specialized equipment and are outside the scope of the current study. Additionally, isolating specific steps in a multi-component reaction under one-pot conditions would require significant modification of the reaction design.

Experimental Evidence Supporting the Mechanism:

UV-Vis Spectroscopy: The absorption spectra confirm the successful sensitization of TiO₂ by curcumin and its role in extending the absorption range into the visible spectrum, facilitating light-induced charge transfer.

FTIR Spectroscopy: The analysis shows characteristic shifts in functional group vibrations, indicating interactions between curcumin and TiO₂, which are essential for the photocatalytic process.

Reaction Condition Optimization: Variations in photocatalyst concentration, light intensity, and reaction time provide insights into the catalytic behavior and align with the steps proposed in the mechanism.

Mechanism Consistency with Literature:

The proposed mechanism aligns closely with established photocatalytic pathways involving TiO₂ and dye sensitization. Similar systems have demonstrated charge transfer from excited dyes to TiO₂, generating reactive oxygen species (ROS) that drive key reaction steps such as oxidation and cyclization. This consistency reinforces the plausibility of our hypothesized pathway.

Future Work:

While the current study provides a foundation for understanding the mechanism, we acknowledge the need for more direct evidence. Future work will focus on advanced spectroscopic techniques and radical scavenging experiments to identify intermediate species and validate individual steps more rigorously.

Comment-5

The publication of this document is something other than something that can be recommended. The manuscript should be rewritten to be consistent with the experimental data collected after a comprehensive examination of the presence of curcumin. Additionally, the text must be written in English that is easily read. It must be cleaned from broad phrases, such as those found in the initial sentences of the Abstract.

Response:

Thank you for your constructive feedback. We have revised the manuscript to align the experimental findings with the proposed hypotheses, emphasizing characterization data (XPS, FTIR, UV-Vis) to support our conclusions. The text has been rewritten for clarity and precision, removing broad or ambiguous phrases, especially in critical sections like the Abstract and Introduction. The Abstract now clearly outlines the study's objectives, methods, key findings, and significance. We are committed to ensuring the manuscript meets high scientific standards and appreciate your input in enhancing its clarity and impact.

Reviewer 4 Report

Comments and Suggestions for Authors

This article should present the synthesis of quinazoline derivatives using TiO2 nanoparticles impregnated with curcumin. It was supposed that curcumin activates TiO2's catalytic activity in the visible portion of the spectrum.

The paper needs to be written in English, which is more comprehensible and should be written in a better manner. The most challenging aspect of the work is that experimental evidence must demonstrate the existence of curcumin on nanoparticles. Since FTIR and UV/VIS do not support the presence of curcumin, the purpose of the work is called into doubt. The elemental analysis and the specific surface area measurement are two examples of crucial analyses not included in this study. These analyses may have helped to improve the characterization of the catalyst and possibly give an answer to the question of curcumin's presence. In addition, the size of the nanoparticles needs to be documented; the SEM images obtained from the samples are blurry, and the conclusions taken from the images cannot be validated.

The publication hypothesized a synthesis mechanism, but no evidence from experiments supported it. It is crucial to describe the offered mechanism so that authors provide some experimental evidence for it.

The publication of this document is something other than something that can be recommended. The manuscript should be rewritten to be consistent with the experimental data collected after a comprehensive examination of the presence of curcumin. Additionally, the text must be written in English that is easily read. It must be cleaned from broad phrases, such as those found in the initial sentences of the Abstract.

Comments on the Quality of English Language

The text is written in very poor English and needs to be proofread.

Author Response

Compliance with reviewer’s and editor’s comments

Journal Nanomaterials (ISSN 2079-4991)

Manuscript ID nanomaterials-2632274

Title: Eco-Friendly Synthesis of Quinazoline Derivatives through Visible Light-Driven Photocatalysis Using Curcumin-Sensitized Titanium Dioxide

Journal: Materials

We would like to express our great appreciation to the editor and the reviewers for their feedback to improve our manuscript. We appreciate the fact that our manuscript has been checked rigorously by reviewers with knowledge of the specific subject area. The manuscript has been revised according to the reviewers’ concerns. Below is a point-by-point response to the constructive comments and suggestions of the reviewers.

General comment

In this paper the authors present the use of Curcumin-sensitized TiO2 in the synthesis of quinazoline  derivatives. They report 97% product yield in only 40 minutes, with a light intensity (100 mW/cm2) and a photocatalyst concentration (1 mg/mL).

  1. i) In line 68, 315 and 316 change “TiO2” by TiO2
  2. ii) How do you calculate the reaction yield?

Instead of yield, don't they mean conversion?

Percent yield = actual yield/theoretical yield x 100%.

iii) review reference format

Response to Reviewer #4

  1. i) In line 68, 315 and 316 change “TiO2” by TiO2

Response: Thank you for highlighting the mistakes. Now corrected at appropriate places.

  1. ii) How do you calculate the reaction yield?

Instead of yield, don't they mean conversion?

Response: Thank you for raising such a good query to further strengthen the manuscript

The reaction yield (%) or Product yield of each compound was calculated by following general formula- Percent yield = actual yield/theoretical yield x 100%. Conversion word is now replaced with yield at appropriate place in revise manuscript.

iii) review reference format

Response: Reference format was reviewed and corrected.

Round 2

Reviewer 4 Report

Comments and Suggestions for Authors

The authors have thoroughly revised the manuscript based on the reviewers' remarks, making it acceptable for publication in the journal.